# Cats Are Not Fish: A Ricker Model Fails to Account for Key Aspects of Trap–Neuter–Return Programs

**DOI:** 10.3390/ani11071928

**Published:** 2021-06-28

**Authors:** Peter J. Wolf, Rachael E. Kreisler, Julie K. Levy

**Affiliations:** 1Best Friends Animal Society, 5001 Angel Canyon Road, Kanab, UT 84741, USA; 2Pathology and Population Medicine, Midwestern University College of Veterinary Medicine, Glendale, AZ 85308, USA; rkreis@midwestern.edu; 3Maddie’s Shelter Medicine Program, University of Florida, Gainesville, FL 32608, USA; levyjk@ufl.edu

**Keywords:** trap–neuter–return (TNR), feral cats, free-roaming cats, community cats, Ricker model

## Abstract

**Simple Summary:**

Various population models have been used to predict the sterilization effort necessary to reduce free-roaming cat numbers through trap–neuter–return (TNR) programs. Among these is a Ricker model, first developed for application in the management of fisheries. We tested this model by using data from two long-term (i.e., >20 years) TNR programs with well-documented population reductions. Doing so revealed that the model cannot account for some key aspects of typical TNR programs. This model is therefore inappropriate for use in assessing the effectiveness of TNR programs. A more recently developed model that accounts for the movement of cats in and out of a given area is better suited for modeling TNR programs.

**Abstract:**

In a frequently cited 2005 paper, a Ricker model was used to assess the effectiveness of trap–neuter–return (TNR) programs for managing free-roaming domestic cat populations. The model (which was originally developed for application in the management of fisheries) used data obtained from two countywide programs, and the results indicated that any population reductions, if they existed, were at best modest. In the present study, we applied the same analysis methods to data from two long-term (i.e., >20 years) TNR programs for which significant population reductions have been documented. Our results revealed that the model cannot account for some key aspects of typical TNR programs, and the wild population swings it predicts do not correspond to the relative stability of free-roaming cat populations. A Ricker model is therefore inappropriate for use in assessing the effectiveness of TNR programs. A more recently developed, stochastic model, which accounts for the movement of cats in and out of a given area, is better suited for predicting the sterilization effort necessary to reduce free-roaming cat numbers through TNR programs.

“*All models are wrong but some are useful*”.George Box

## 1. Introduction

The use of trap–neuter–return (TNR) as a humane alternative to the lethal removal of free-roaming domestic cats originated in Europe in the 1950s and was adopted in the US beginning in the early 1990s [1]. For many years, the practice received relatively little attention from researchers interested in evaluating its effectiveness, resulting in what some have described as an “information vacuum” [2]. More recently, however, several studies have been published on the subject. Some have reported the results of targeted TNR programs producing significant long-term population reductions [3,4,5,6,7], while others have documented significant reductions in feline intake and euthanasia at animal shelters where high-intensity targeted TNR has been implemented [8]. Still others have reported reductions in feline intake and euthanasia resulting from shelter-based TNR programs [9] and programs integrating community-based and shelter-based TNR efforts [10,11].

By contrast, studies of TNR programs implemented over just one [12] or two years [13] have documented population increases, similar to the short-term increases documented following the implementation of TNR in one long-term study [14]. A recent study of outdoor cats in Stillwater, Oklahoma, USA, used trail cameras to document a reduction from 47 to 35 cats (25.5%) or, “after correcting for detectability,” from 62 cats to 48 cats (22.6%) over 5 years [15], exceeding those predicted by stochastic simulation modeling (15.5%) for a scenario in which 40% of the unsterilized population are sterilized every 6 months for 5 years [16].

Although some have suggested that the removal of cats and kittens—especially for the purposes of adoption—is somehow separate from, or in addition to, TNR efforts (see, for example, [17,18]), removal has been integral to TNR from its inception, an obvious “win” both in terms of immediate population reductions and animal welfare concerns [19], as cited in [1]. In populated areas, TNR invariably attracts removal and humane euthanasia, but to varying degrees. TNR on an uninhabited island, on the other hand, would likely have different results, both because of less removal and euthanasia and less immigration. Throughout this paper, we use the term “TNR” to refer to sterilization programs that incorporate the regular removal of cats and kittens (mostly for adoption, but also, when appropriate, for euthanasia).

### 1.1. Modeling Free-Roaming Cat Population Dynamics

In addition to the empirical studies examining the effectiveness of TNR, at least three different types of population model have been developed. Using a matrix population model to “explore how cat populations may respond to various forms of control,” Andersen et al. [20] concluded that “control strategies that target survival of free-roaming cats should be more effective at reducing cat populations than those that target fecundity.” Specifically, the authors found population reductions could be achieved by sterilization only if more than 75% of the fertile population was sterilized annually. Similar reductions could be achieved by the lethal removal of at least 50% of the population. The City of Los Angeles recently used a matrix population model to estimate the results of a program to sterilize 20,000 free-roaming cats annually for 30 years, concluding that this program “would result in fewer free-roaming cats throughout the City and a lower rate of population growth due to reduced reproductive output” compared to a scenario in which the 20,000 annual sterilizations were not provided [21].

Foley et al. [22] combined empirical data with a theoretical population model “to describe population dynamics of the feral cats and modifications to the dynamics that occurred as a result of… TNR programs”. The authors used data obtained from two large-scale TNR programs (San Diego County, California, and Alachua County, Florida) as input values for a Ricker population model [23], first developed for the management of fisheries, and found “no indications of a significant reduction in per capita growth rate” in either program evaluated. Moreover, the data revealed no reductions in the proportion of pregnant female cats sterilized as part of the two programs.

Miller et al. [16] used a “stochastic demographic simulation approach to evaluate” various free-roaming cat management schemes in the first population modeling to account for cats moving in (e.g., via abandonment) and out (e.g., via dispersal) of a given area. Results showed that, in large urban areas, free-roaming cat populations can be reduced by removing 20% of the population every 6 months or by sterilizing 30% of the fertile component of the population every 6 months. Building on this work, Boone et al. [24] concluded that the intensity of management—whether removal or sterilization—“is important not only to reduce populations more quickly, but also to minimize the number of preventable deaths that occur over time”.

### 1.2. References to the 2005 Study in the Literature

In the years since Foley et al. [22] was first published, some critics of TNR have cited the analysis to support the claim that TNR is ineffective at reducing the population of free-roaming cats [18,25,26,27,28]. Others go further, arguing that the study is evidence of population increases despite long-term TNR efforts [17,29,30,31].

In the present study, we compare the results from Foley et al. [22] to those from two long-term TNR studies documenting significant population reductions.

## 2. Materials and Methods

To assess the effectiveness of TNR to reduce free-roaming cat populations, Foley et al. [22] examined data obtained from two large-scale TNR programs: San Diego County, California, USA (1992–2003, *n* = 14,452), and Alachua County, Florida, USA (1998–2004, *n* = 11,822). As free-roaming cat population data were unavailable, the authors used annual surgery data as an index of each community’s free-roaming cat population. Specifically, Foley et al. [22] examined these datasets by applying the following four criteria:(1)Density-dependent population regulation, as determined by regression analysis of per capita growth rate over time;(2)Changes in fecundity, as determined by regression analysis of the proportion of pregnant female cats over time;(3)The calculated Malthusian multiplier (which “must be <1.0 for the population to be in decline”); and(4)The proportion of sterilized cats (relative to the overall population) that is necessary to decrease the population.

Foley et al. [22] used a Ricker model to predict “population regulation” based on the following formula:
$$R_{t}=e^{r_{m}\left(1-\frac{N_{t}}{K}\right)}$$
“where R_t_ is an annual population multiplier or net fundamental reproductive rate, r_m_ is the maximum per capita rate of increase, N_t_ is the population size at time t, and K is the carrying capacity. If R_t_ = 1, the net annual growth of the population, r_t_, is 0 (i.e., the population size is multiplied by 1.0)”.

The critical neutering rate, s, was calculated as follows:
s = (R_m_ − 1)/(R_m_ − p)
where
$$R_{m}=e^{r_{m}}$$
and p is the annual survival rate. Here, Foley et al. [22] used a value of 0.8, based on a mean lifespan of 5 years.

The annual neutering fraction necessary to produce a population reduction, s_a_, was then calculated as follows:
s_a_ = s(1 − p)

For the present study, we first attempted to replicate the results presented by Foley et al. [22] and then, having done so, tested the authors’ model against the results of two long-term studies demonstrating significant reductions in free-roaming cat populations [3,32]. As Foley et al. [22] lacked detailed information about the free-roaming cat populations examined, the number of sterilization surgeries conducted each year was used as an index of the countywide free-roaming cat population. For the present study, we used detailed census data as model parameters. The first dataset was from a long-term TNR program (known as ORCAT) in a private community of Key Largo, Florida, USA, where Kreisler et al. [3] reported a 55% reduction, from 455 cats to 206, over 14 years. This result was based on the analysis of 10 censuses conducted by a paid caregiver who “was highly knowledgeable of the entire [free-roaming] population, which she interacted with on a daily basis” [3]. The second dataset we used was from the University of Central Florida (UCF) campus TNR program, where the population was reduced from 68 cats to 10 (85%) over 28 years. Here, too, the result was based on census data, in this case records from “daily feeding sessions” when “absences and new arrivals” were observed by caregivers [32].

Statistics were performed using the Stata software package, v.17. Statistical significance was expressed using *p*-values and *α* = 0.05. Ricker model simulations were performed for a period of 20 years using the R programming language, with input values for R, K, and starting populations provided either by Foley et al. [22] (for San Diego and Alachua counties) or calculated for the ORCAT and UCF data (based on annual surgeries, following the methods described by Foley et al. [22]).

## 3. Results

### 3.1. Discrepancies Found in the Original Study

Annual surgery data were approximated based largely on the plots provided by Foley et al.’s Figure 1 in [22] and supplemented with discrete values referenced in the text (e.g., during 2003, “the total numbers of trapped cats were 1514… in San Diego County”). This dataset was then used to calculate annual per capita growth rates (Table 1). Following Foley et al. [22], the annual growth rate, r_t_, was calculated according to the following formula:
r_t_ = ln(N_t+1_/N_t_)
where N_t_ represents the annual surgeries in a given year. For example, San Diego County’s annual growth rate for 1993 was calculated as follows:
r_1993_ = ln(1630/1768) = −0.08

Other annual per capita growth rates were calculated in the same manner. Doing so revealed two discrepancies for San Diego County data: the calculated values (based on annual surgery values obtained from Foley et al.’s Figure 1) do not correspond with the values illustrated in their Figures 4 and 5 for 1994 (−0.18 from Figure 1 vs. −0.39 from Figures 4 and 5) and 1998 (−0.56 vs. −0.30).

For our analysis, we used the calculated values of per capita growth rate shown in Table 1. We also discovered that the annual surgery data presented in Figure 5A from Foley et al. [22] (San Diego County) do not match the corresponding data presented in their Figure 1A. For example, Figure 1A from Foley et al. [22] indicates that only once, in 1993, did annual surgeries exceed 1750; yet, four of the eight points plotted in their Figure 5A correspond to annual surgeries exceeding 1750. Moreover, Figure 5A includes only eight points despite there being 12 years of available data. These discrepancies are illustrated in Figure 1 below.

Finally, the measure of statistical significance (i.e., *p*-value) reported by Foley et al. [22] for the linear regression of per capita growth rate over time in Alachua County (*p* = 0.10) is quite different from the result we obtained from the same analysis (*p* = 0.50). Additionally, the value reported by Foley et al. [22] for the linear regression of per capita growth rate vs. annual surgeries for San Diego County (*p* = 0.09) is slightly higher than our result (*p* = 0.04), likely a consequence of assessing only 8 of the 10 available data points and/or the aforementioned discrepancy between the data shown in Figures 1A and 5A. Although these discrepancies change the overall conclusions reported by Foley et al. [22] only slightly, the relationship of San Diego County’s per capita growth rate to annual surgeries is statistically significant in our analysis. In any case, we have used the values shown in Table 1 when referencing specific results.

### 3.2. Per Capita Growth Rate

Per capita growth rate over the 23 years for which ORCAT data were available ranged from −0.99 to 0.96, with a value of −0.32 in 2016, the final year for which such a calculation was possible (Figure 2a). Regressing per capita growth rate over time revealed no significant reduction in per capita growth rate (*p* = 0.52). Per capita growth rate over the 25 years for which UCF data were available ranged from −1.29 to 1.79, with a value of −1.10 in 2018, the final year for which such a calculation was possible (Figure 2b). Regressing per capita growth rate over time revealed no significant reduction in per capita growth rate (*p* = 0.98).

Regressing per capita growth rate against annual sterilization surgeries for the ORCAT program revealed a significant decline (*p* = 0.03). Two outliers that were identified via residual analysis were excluded to improve linearity; this increased the overall significance (*p* = 0.003). The maximum per capita rate of increase (corresponding to the y-intercept) was 0.89 and the estimated index carrying capacity (corresponding to the x-intercept) was approximately 62 cats (Figure 3a). Regressing per capita growth rate against annual sterilization surgeries for the UCF program revealed no significant change (*p* = 0.11). The maximum per capita rate of increase was 0.16 and the estimated index carrying capacity was 3.4 cats (Figure 3b).

LOWESS smoothing indicated a non-linear relationship in the data, particularly for the UCF data (Figure 4).

Foley et al. [22] used annual surgeries as an index of free-roaming cat populations “because actual population counts were not available or practical.” Both the ORCAT and UCF programs implemented regular censuses, however, allowing us to examine the relationship between per capita growth rates and free-roaming cat populations directly (Figure 5). Analysis of the residuals associated with the ORCAT data led us to consider one data point (from 2016) an outlier; regression of the remaining data revealed no significant association of per capita growth rate over time (*p* = 0.73). (Removal of the data point from 2016 had little effect on the regression analysis.) Regression of the UCF data revealed no significant relationship (*p* = 0.09).

### 3.3. Proportion of Pregnant Female Cats

Regressing the proportion of pregnant female cats (relative to all female cats regardless of age) over the 23 years for which ORCAT data were available (Figure 6) revealed a significant decrease (*p* < 0.0001). Unfortunately, comparable data from the UCF program were unavailable.

### 3.4. Malthusian Multiplier

The value of the Malthusian multiplier, Rm, was derived using the formula:$$R_{m}=e^{r_{m}}$$
where r_m_ is the maximum per capita rate of growth (corresponding to the y-intercept in Figure 3). Following Foley et al. [22], we used the number of cats sterilized annually as an index of the larger free-roaming cat population; the resulting value of R_m_ for the ORCAT program was 2.45. The estimated index carrying capacity (which corresponds to the x-intercept) was 62 cats. The actual carrying capacity was estimated (again, following the methods of the original analysis) by multiplying the index carrying capacity by the free-roaming cat population (during the final year of the program) and then dividing by the number of surgeries (during the final year of the program). Doing so resulted in an estimated carrying capacity of 101 cats for the ORCAT program. However, Kreisler et al. [3] documented an initial population of 455 cats at this site, more than seven times this estimate. For the UCF program, R_m_ = 1.17, the estimated index carrying capacity was 3.4 cats, and the actual carrying capacity was estimated to be 34 cats. However, the initial census of UCF cats suggested that there were 68 cats on campus at the time [33].

### 3.5. Critical Neutering Rates

Again following Foley et al. [22], critical neutering rates were derived from values of R_m_ and annual survival rates (p). Specifically, the critical neutering rate, s, was calculated as follows:s = (R_m_ − 1)/(R_m_ − p)
where
$$R_{m}=e^{r_{m}}$$
and p is the annual survival rate.

Annual survival rates were in turn estimated from median lifespan as follows:
p = 1 − (1/median lifespan)

Table 1 in Foley et al. [22] shows values of critical neutering rates for various values of R_m_ and p.

The median lifespan for the ORCAT program was 6 years; therefore, the critical neutering fraction, s, was:s = (2.45 − 1)/(2.45 − 0.83) = 90%

Again, following Foley et al. [22], the annual neutering fraction necessary to produce a population reduction, s_a_, was calculated as follows:s_a_ = s(1 − p)

Table 2 in Foley et al. [22] shows values of critical annual neutering rates for various values of R_m_ and p.

Given the median lifespan (6 years) and annual neutering fraction (90%) for the ORCAT program, the critical annual neutering fraction, s_a_, was:
= 0.90(1 − 0.83) = 15%

Following the same analysis, the critical neutering fraction for the UCF program was 41% and the critical annual neutering rate was 10%, based on an estimated median lifespan of 4 years (unpublished data).

As Foley et al. [22] point out, “survivorship may differ between neutered and non-neutered cats.” Indeed, an examination of the ORCAT program found that the “mean age of cats at removal increased 1.9 months per year over time (*p* < 0.0001) from 6.4 months in 1995 to 77.3 months in 2017” [3].

The actual annual neutering fractions were estimated by dividing the number of sterilizations by the number of cats in the population, as determined by linear regression analysis of periodic census data (for the ORCAT data) or periodic census data (for the UCF data). The median annual neutering fraction for ORCAT was 0.22, with a low of 0.08 in 2004 and a high of 0.61 in 2017. The median neutering fraction for UCF was 0.16, with a low of 0 in 2008 and a high of 0.43 in 2015 (Figure 7).

### 3.6. Ricker Simulations

The Ricker equation used by Foley et al. [22] was used to run simulations for San Diego County, Alachua County, ORCAT, and UCF (Table 2 and Figure 8). The ORCAT and Alachua County simulations revealed a large variation in projected cat populations year to year as the number of cats exceeded the calculated carrying capacity, and the cat population was predicted to collapse, rebound, then collapse again. The San Diego and UCF simulations initially oscillated around the calculated carrying capacity before settling at the carrying capacity.

## 4. Discussion

To “evaluate development and implementation of models that could determine program success and calculate the rate of neutering needed to decrease the feral cat population,” Foley et al. [22] used data from two large-scale TNR programs. In their analysis, the authors focused on four criteria in particular: (1) density-dependent population regulation; (2) changes in fecundity; (3) the Malthusian multiplier; and (4) critical neutering rates. In the present study, we applied the same analysis to datasets of individual cat records obtained from two long-term (i.e., >20 years) TNR studies in which significant population declines were documented [3,32]. Our results revealed that the model cannot account for some key aspects of typical TNR programs (Table 3).

Specifically: (1) our regression analysis using the Ricker model of per capita growth rate over time predicted no statistically significant population reduction for either the ORCAT or UCF program—in direct conflict with the observed progressive population reductions achieved by those programs. More importantly, our analysis showed that the observed field data were non-linear, raising questions about the use of linear regression; (2) the proportion of pregnant cats from the ORCAT program decreased significantly over time (*p* < 0.0001), although no comparable analysis was possible for the UCF program due to the lack of data; (3) neither Malthusian multiplier calculated was less than 1.0, although each program documented significant population declines; (4) and the median annual neutering rates for both programs (0.22 for ORCAT and 0.16 for UCF) exceeded the necessary critical annual neutering rates (0.15 for ORCAT and 0.10 for UCF). These mixed results suggest that the analysis described by Foley et al. [22] to assess the effectiveness of a typical TNR program (i.e., one in which kittens and sociable senior cats are often pulled for adoption) or of TNR to reduce free-roaming cat populations more generally [17,18,28,30,31] overlooks the complexities inherent in such management programs.

To be clear, Foley et al. [22] were more cautious about their findings than is sometimes acknowledged by those citing their study. The authors describe their results as “mixed… regarding the success of large TNR programs in San Diego and Alachua counties,” for example, and that “any population-level effects were minimal” [22]. As noted previously, however, a number of subsequent references to the work suggest far more definitive (negative) conclusions. Below, we discuss some of the factors that likely contributed to the discrepancies between the model results and the empirical evidence.

### 4.1. Using Annual Surgeries as an Index for Free-Roaming Cat Populations

Foley et al. [22] used annual surgeries as an index for free-roaming cat populations, arguing that “the trajectories of populations (whether or not populations were declining) could be determined from calculation of maximum per capita rate of increase without accurately detecting population size or carrying capacity”. However, the number of surgical appointments in a given year can—and often does—vary considerably for reasons having nothing to do with population size or carrying capacity. Grant funding, for example, is often based on a number of factors independent of such data (e.g., support from shelter staff and policymakers). Surgical capacity can often be a barrier, as shelters and low-cost clinics struggle to recruit and retain qualified veterinary professionals. Moreover, robust TNR programs typically rely on a sufficient number of volunteers for trapping, transport, recovery, and any number of other critical tasks. To use annual surgeries as a proxy for free-roaming cat populations—especially across entire counties—is therefore problematic.

Commenting on their results, Foley et al. [22] observed:

“The regression of per capita growth rate on population size was not significant for either San Diego or Alachua counties, possibly reducing confidence in the estimate of population growth rates. However, this was not surprising given that *a time series of at least 20 years is typically required before such a regression is found to be significant*” (emphasis added).

As noted above, per capita growth rate was, in fact, found to be negatively associated with San Diego County’s annual surgeries (as an index of population size) at a statistically significant level (*p* = 0.04). Both the ORCAT and UCF datasets exceeded the 20-year requirement suggested; regression of the ORCAT data revealed a significant negative association between per capita growth rate and annual surgeries (*p* = 0.003), while regression of the comparable UCF data revealed no significant relationship (*p* = 0.11). It is clear, therefore, that this metric is not a reliable measure of population reductions. Moreover, analysis of the data using LOWESS smoothing illustrates the non-linear relationship of the data (Figure 4), raising additional questions about the use of linear regression. Linear regression of the per capita growth rates as a function of free-roaming cat populations also revealed no significant relationships, suggesting that this metric, too, is not a reliable measure of population reductions.

It is notable that the carrying capacities estimated based on annual surgeries for all programs were considerably less than those estimated based on households feeding cats (San Diego and Alachua counties) or population censuses (ORCAT and UCF). This further supports our observation that annual surgeries are not an accurate proxy for free-roaming cat populations.

### 4.2. Accounting for Cats Removed for Adoption

Boone [34] observed that “ancillary techniques, such as the removal of socialized cats and socializable kittens for adoption, will complement TNR efforts, and may be critical in achieving population size reduction.” However, the modeling used by Foley et al. [22] was unable to account for cats removed for adoption, a common component of TNR programs [4,5,33]. In both the ORCAT and UCF programs, adoptions accounted for a significant number of cats and kittens removed from the population and therefore are not included in census counts (Figure 9). It is unclear from Foley et al. [22] whether such adoptions from the San Diego and Alachua county programs occurred and, if so, were tracked. It is worth noting, however, that approximately 21% of pet cats in the US are obtained directly from the “stray” population [35]. Another 21% are obtained from friends and relatives and 10% more from rescue groups, sources that are likely to remove cats from the “stray” population [35]. Although it is possible that a significant—albeit unknown—number of cats who were sterilized as part of the San Diego and Alachua county TNR programs were later adopted as pets, such adoptions were unlikely to reach the high levels observed in the ORCAT and UCF programs.

Although both the ORCAT and UCF programs removed a considerable number of cats and kittens for adoption, it is worth noting that the median age of cats removed for adoption from the ORCAT program was 4.8 months. This suggests that they were predominantly kittens and therefore among the “doomed surplus,” 75% of which were unlikely to survive to 6 months of age [36]. Data from the UCF program also indicated that kittens were most likely to be adopted.

### 4.3. The Difficulties Associated with Evaluating TNR across Large Geographic Areas

Foley et al. [22] note that the two TNR programs they examined “probably were performed on too large a scale; many cats were neutered, but this constituted a very small overall proportion of the cats. Moreover, feral cats within a county surely do not constitute a single population, further diluting the enormous overall effort into numerous smaller efforts with less impact”. Indeed, attempting to measure reductions in free-roaming cat populations across large areas is difficult under any circumstances; to make such an attempt using annual surgeries is even more so. In 2000, the human population of San Diego County (with 10,895 km^2^ land area [37]) was approximately 2.81 million [38], while the population of Alachua County (2266 km^2^ land area [39]) was approximately 218,000 [38]. It is unclear from Foley et al. [22] whether the two TNR programs examined served their entire county, and whether their service areas might have changed over time. In any case, it is not surprising that significant population reductions were not detected across such large, densely populated areas. There is no way to know, from the information provided, how clinic appointments were distributed across each program’s service area. Sometimes this is performed simply on a first-come-first-served basis; in other cases, programs target “hot spots”—those parts of a service area where there are known to be a large number of cats. Grant funding is often targeted at such hot spots, thus giving priority to cats in these areas. Additionally, some TNR programs emphasize the importance of sterilizing as many cats in a given area (e.g., city block, neighborhood) as possible before moving on to other parts of their service areas.

It is not at all unreasonable to assume that sterilization rates across both San Diego County and Alachua County varied considerably, with some groups of cats well managed (i.e., their numbers stabilized and then decreasing) while others were managed less rigorously or unmanaged entirely. Variability in sterilization rates occurs over time as well: hot spots targeted with sufficient intensity “cool off,” allowing resources to be allocated elsewhere. Aggregate data, such as those used by Foley et al. [22], obscure these important aspects of free-roaming cat management, making it virtually impossible to accurately assess the effectiveness of TNR at a more localized scale (e.g., neighborhood, census block, zip code). The challenges associated with the modifiable areal unit problem (e.g., the results of analyses varying considerably depending on which boundaries are used) are well documented [40].

### 4.4. The Importance of Targeting TNR Efforts for Population Reduction

Foley et al. [22] recommend that TNR programs “should be focused on well-defined, preferably geographically restricted, cat populations, rather than diluting effort across multiple populations.” The benefits of such targeting have since been demonstrated with population modeling. Building on previous work [16], Boone et al. [24] used a stochastic simulation model to evaluate various methods of managing free-roaming cat populations (e.g., no action, removal, culling, and TNR). Model results revealed the importance of intensity for both population reductions and minimizing “preventable deaths” (i.e., cats killed by lethal removal and kittens dying before adulthood), prompting the authors to warn: “With sufficient intensity, management by TNR offers significant advantages in terms of combined lifesaving and population size reduction. At lower intensity levels, these advantages are greatly reduced or eliminated” [24]. Given limited resources, high intensity requires a targeted approach to sterilization efforts. The Ricker model used by Foley et al. [22], however, has no way to account for such targeting. A program delivering 100 surgeries annually across an entire city, for example, is seen as no more “effective” than one delivering 100 surgeries annually to a neighborhood in greatest need. As the modeling work from Boone et al. [24] demonstrates, such oversimplifications overlook two key objectives of TNR programs: reducing both free-roaming cat populations and unnecessary mortalities.

### 4.5. Calculated Sterilization Rates and the Feasibility of TNR

Foley et al. [22] reported critical neutering rates (71–94%) “far greater than what was actually achieved”, a point subsequently used by some critics of TNR to argue that the management scheme is infeasible [17,29]. It is important to note, however, that this is a cumulative sterilization rate—typically the result of TNR efforts spanning several years. Such high rates are likely to be seen as especially unrealistic in the context of countywide free-roaming cat populations estimated to be in the tens or hundreds of thousands. It may be more appropriate to focus on critical annual neutering rates because they better reflect typical TNR efforts. We documented annual neutering rates for both programs that regularly exceeded the critical threshold necessary to produce a population reduction; indeed, both programs resulted in significant reductions over time.

More recent modeling work—which, unlike a Ricker model, accounts for cats moving in and out of a given area—found that in large urban areas free-roaming cat populations can be reduced by removing 20% of the population every 6 months or by sterilizing 30% of the fertile component of the population every 6 months [16]. Although Foley et al. [22] described a 75% sterilization rate as “unrealistic,” the modeling results from Miller et al. [16] show that sterilizing 40% of the intact portion of the population every 6 months for 10 years will achieve a 75% sterilization rate. Moreover, several studies have shown that cats tend to congregate in groups of 3–12 cats [3,5,7,10,14,41,42]. Seen in this context (e.g., sterilizing at least 3 cats in a group of 10 during the first 6 months of effort), TNR is likely to be seen as considerably more feasible for managing free-roaming cats.

### 4.6. Health Benefits Associated with TNR

Although the focus of the modeling by Foley et al. [22] was potential population reductions, it is important to note that there are health benefits to TNR as well. At least one study has documented improved body condition scores (BCS) following the sterilization of free-roaming cats, for example [43]. Another study noted improved BCS for both intact and sterilized cats observed in the same area, prompting the researchers to suggest that perhaps “less aggressive behavior may result in a reduction of competitive behavior by neutered cats, which may enable other cats to gain additional access to vital resources” [44]. Another study demonstrated that cats vaccinated against rabies, an important public health threat, and against feline infectious diseases during the TNR process had a robust immune response, protecting not only the treated individuals but also enhancing herd immunity in the community [45]. A study of more than 100,000 cats admitted to TNR programs across the US documented that, while untreatable health conditions were rare (0.4% of cats admitted), TNR programs also served as a provider of humane euthanasia, an important component of animal welfare [46].

Studies of TNR programs have also documented reductions in the number of free-roaming cats reported dead [9,10], perhaps, as had been suggested by other researchers, because neutered male cats are less likely to roam [44]. Data from the ORCAT program revealed declining rates for both feline immunodeficiency virus (FIV, 0.16% annually, *p* = 0.013) and feline leukemia virus (FeLV, 0.18% annually, *p* = 0.033) [3]. Such findings correspond well with those from other studies demonstrating the beneficial effects of sterilization on reducing the transmission of both diseases: sexually intact male cats are prone to a higher rate of FIV infection and kittens are more susceptible than adults to FeLV infection [47,48,49].

### 4.7. Ricker Simulations

The Ricker simulations did not follow the patterns observed in either of the locations for which population data from the field were available (ORCAT and UCF). In the case of ORCAT, the actual population (starting at 636% of estimated carrying capacity using the method described by Foley et al. [22]) declined linearly over time and was not subject to the dramatic increases and decreases predicted by the model. In the case of UCF, the population (starting at 200% of the model’s estimated carrying capacity) declined linearly over time before settling in at a steady population (10) well below estimated carrying capacity (30% of estimated carrying capacity). While actual population data were not available for San Diego County or Alachua County, the carrying capacity for San Diego was estimated to be 210,325 as compared to the estimated population of 240,690 (population 114% of carrying capacity), while the carrying capacity was estimated to be 19,323 for Alachua County as compared to the estimated population of 36,398 (population 188% of carrying capacity) [22]. The dramatic cycles of increase and collapse in simulated populations were predicted for locations with the largest differences between free-roaming cat populations and estimated carrying capacity (i.e., Alachua County and ORCAT—see Figure 8). These dramatic boom–bust cycles—the impetus for Ricker’s [23] interest in developing models applicable to the management of fisheries—have not been documented for cat populations, which may be because cat populations are more appropriately modeled via contest competition models, where available resources are utilized differentially by individuals, rather than scramble competition models (such as a Ricker model), where available resources are shared equally.

## 5. Study Limitations

Our study includes some notable strengths over Foley et al. [22], including datasets spanning roughly twice the duration of those used in the original analysis and free-roaming cat population censuses far more accurate than could be expected from countywide estimates. Nevertheless, it is worth pointing out that we lacked the data necessary to precisely replicate the results reported in the original paper; we instead approximated the data based on the plots provided as well as discrete values referenced in the text. In addition, the lack of UCF data documenting pregnant cats limited our analysis of that particular metric to the ORCAT data.

## 6. Conclusions

To evaluate the validity of the Ricker model used by Foley et al. [22] to assess the effectiveness of TNR programs, we used the relevant data from two long-term (i.e., >20 years) TNR programs for which significant population reductions have been documented [3,32]. The mixed results we observed for the four criteria based on the Ricker model demonstrated that this model cannot account for some key aspects of typical TNR programs (e.g., cats and kittens removed for adoption). Moreover, the model’s wild population swings do not correspond to the relative stability of the free-roaming cat populations documented here (and elsewhere). Therefore, a Ricker model should not be considered a valid model for assessing the effectiveness of TNR programs.

## Figures and Tables

**Figure 1 animals-11-01928-f001:**
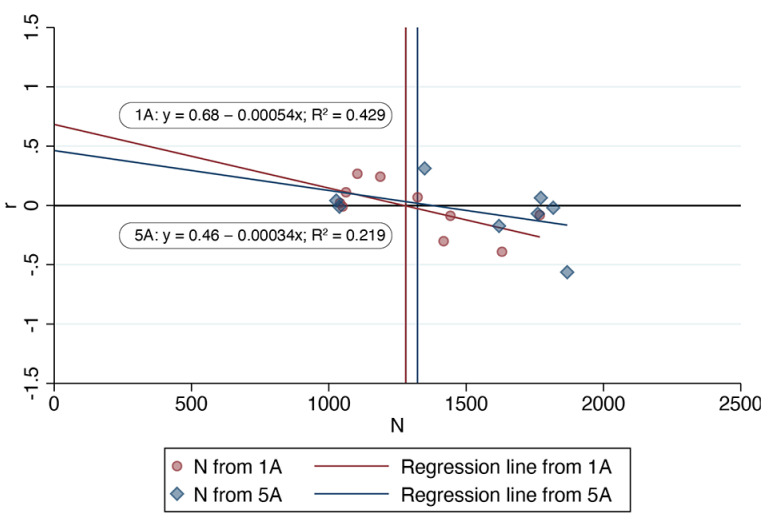
Calculated per capita growth rate vs. annual sterilization surgeries, San Diego County 1993–2002, estimated from Foley et al. [22] Figures 1A and 5A.

**Figure 2 animals-11-01928-f002:**
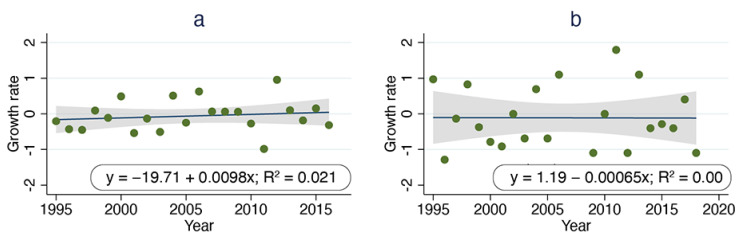
Calculated per capita growth rate over time, ORCAT program 1995–2016 (**a**) and UCF program 1995–2018 (**b**). Dark blue lines are best-fit lines derived from linear regression.

**Figure 3 animals-11-01928-f003:**
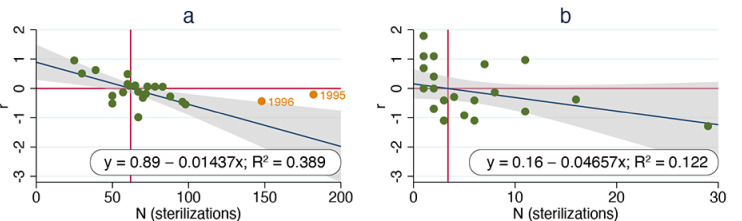
Calculated per capita growth rate vs. annual sterilization surgeries, ORCAT program 1995–2016 (**a**) and UCF program 1995–2018 (**b**). Dark blue lines are best-fit lines derived from linear regression; vertical red lines correspond to x-intercepts. Outliers (as determined via residual analysis) for ORCAT data are shown in orange and excluded from linear best-fit.

**Figure 4 animals-11-01928-f004:**
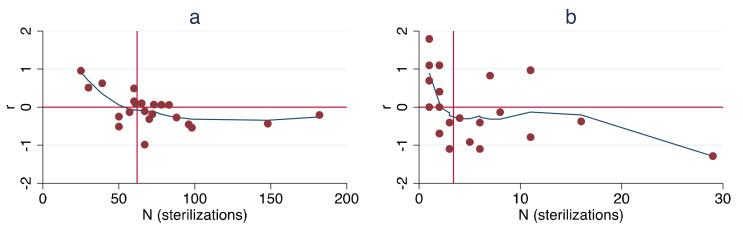
Calculated per capita growth rate vs. annual sterilization surgeries with LOWESS smoothing, ORCAT program 1995–2016 (**a**) and UCF program 1995–2018 (**b**).

**Figure 5 animals-11-01928-f005:**
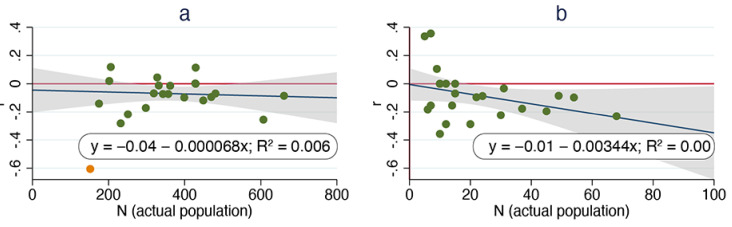
Calculated per capita growth rate vs. free-roaming cat population, ORCAT program 1995–2016 (**a**) and UCF program 1995–2018 (**b**). Dark blue lines are best-fit lines derived from linear regression. Outlier (as determined via residual analysis) for ORCAT data is shown in orange and excluded from linear best-fit.

**Figure 6 animals-11-01928-f006:**
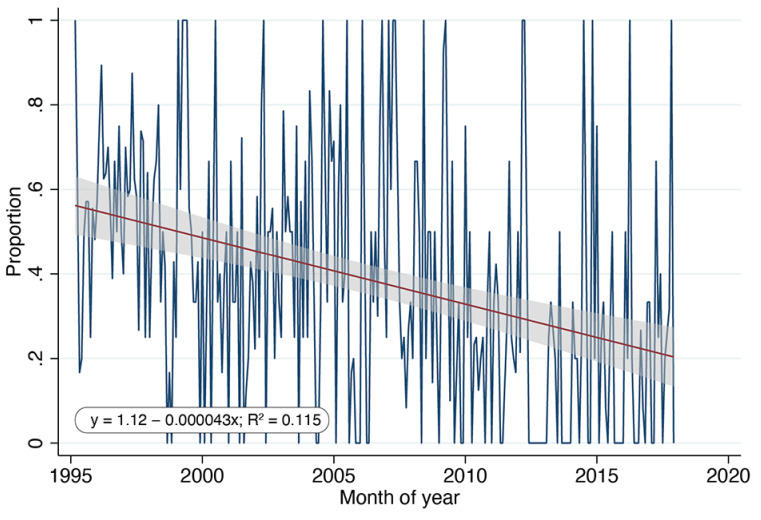
Proportion of female cats relative to all female cats, regardless of age, determined to be pregnant at time of surgery each month, ORCAT program (1995–2016). Comparable data from the UCF program were unavailable.

**Figure 7 animals-11-01928-f007:**
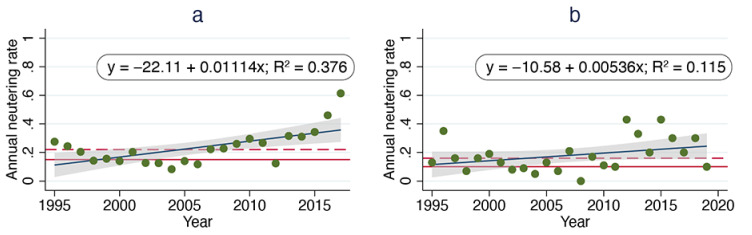
Observed annual neutering rates over time, ORCAT program 1995–2016 (**a**) and UCF program 1995–2018 (**b**). Dark blue lines are best-fit lines derived from linear regression; solid red lines represent critical annual neutering rates; broken red lines represent median annual neutering rates.

**Figure 8 animals-11-01928-f008:**
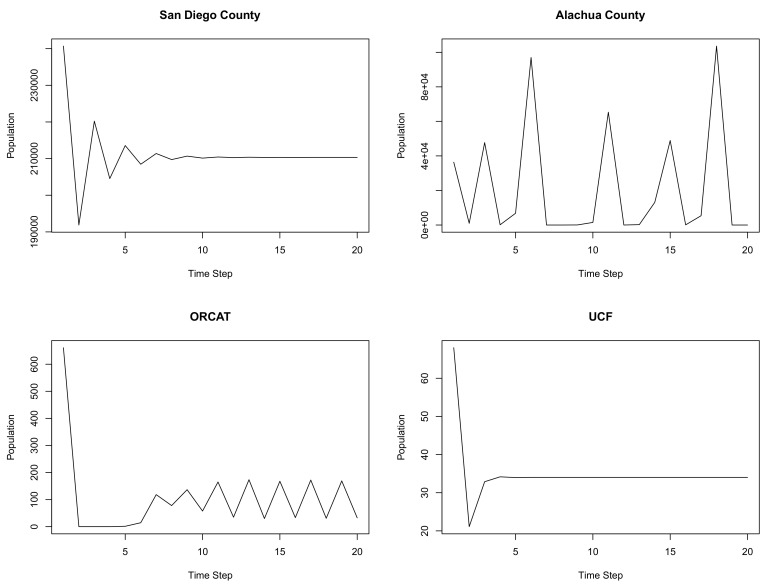
Ricker model simulation for San Diego and Alachua counties (**top**) and ORCAT and UCF programs (**bottom**). Time step is years.

**Figure 9 animals-11-01928-f009:**
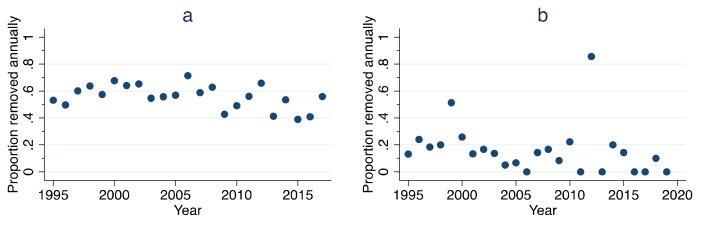
Proportion of cats removed annually for adoption, ORCAT program 1995–2016 (**a**) and UCF program 1995–2018 (**b**).

**Table 1 animals-11-01928-t001:** Estimated annual surgeries and corresponding annual growth rates adapted from Figure 1 of Foley et al. [22]. As per the original analysis, annual growth rates were not calculated for the first year of each program.

Year	San Diego County		Alachua County	
	Annual Surgeries	Annual Growth Rate	Annual Surgeries	Annual Growth Rate
1992	81	–	N/A	N/A
1993	1768	−0.08	N/A	N/A
1994	1630	−0.18	N/A	N/A
1995	1104	0.30	N/A	N/A
1996	1443	−0.02	N/A	N/A
1997	1323	0.06	N/A	N/A
1998	1418	−0.56	666	–
1999	1050	−0.01	1641	−0.11
2000	1041	0.03	1468	0.45
2001	1062	0.12	2273	−0.20
2002	1188	0.25	1852	0.18
2003	1514	−0.08	2213	−0.39
2004	N/A	N/A	1480	−0.11

**Table 2 animals-11-01928-t002:** Values used for Ricker model simulations. Data for San Diego and Alachua counties are from Foley et al. [22]; values for ORCAT and UCF are from the present study.

Location	R_m_	K	N_1_
San Diego	1.57	210,325	240,690
Alachua	4.1	19,323	36,398
ORCAT	2.45	101	661
UCF	1.17	34	68

**Table 3 animals-11-01928-t003:** Key metrics, as described by Foley et al. [22], applied to ORCAT and UCF TNR programs. Results from the original study (San Diego and Alachua counties) are included for reference (data were approximated based on the plots provided in the original publication—see Figure 1).

	San Diego County (California, USA)	Alachua County (Florida, USA)	ORCAT	UCF
Per capita growth rate over time	not significant (*p* = 0.21)	not significant (*p* = 0.50) *	not significant (*p* = 0.52)	not significant (*p* = 0.98)
Per capita growth vs. annual surgeries	significant decline (*p* = 0.04) *	not significant (*p* = 0.10)	significant decline (*p* = 0.003)	not significant (*p* = 0.11)
Per capita growth vs. population	N/A	N/A	not significant (*p* = 0.73)	not significant (*p* = 0.09)
Proportion of female cats pregnant	not significant (*p* = 0.87)	not significant (*p* = 0.95)	significant decline (*p* < 0.0001)	N/A
Malthusian multiplier, Rm	1.98 *	4.10	2.45	1.17
Critical overall neutering rate	0.71	0.94	0.88	0.41
Critical annual neutering rate	0.14	0.19	0.15	0.10

* These values correspond to the data provided by Foley et al. [22] although the text indicates different values (see Section 3.1 for details).

## Data Availability

The data presented in this study are available upon request.

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
