# Peer review of "Cats Are Not Fish: A Ricker Model Fails to Account for Key Aspects of Trap–Neuter–Return Programs"

_animals, 2021, doi:10.3390/ani11071928_

Round 1
Reviewer 1 Report
This is a fascinating manuscript that closely examines some assumptions and prior literature about cat modelling. Given the relative dearth of actual data which has been modelled and the variety of models used, this adds some real substance to the literature. Many of my comments are a bit nit-picky but for this type of analysis, I think that is appropriate.
One comment to consider throughout: we tend to talk about “population declines” as the same end point as “population stabilization” or zero growth rates. These two situations can require quite different intensity of interventions—so we need to be thoughtful about when we are looking for R1=0 vs R1 a little or a lot less than zero. I think some of the controversy about how much TNR is needed can be linked to this lack of clarity. I would also note early in the manuscript that these new programs presented here are not “just TNR” they include adoption and removal. While I agree that TNR can and likely should be used as a generic term for an approach including trapping, sterilization and return, some authors have argued that the inclusion of removal means that the effort isn’t TNR anymore.
Line 100: for those who are less familiar with Foley et al and some of that terminology can you please edit this bullet (or add some additional explanation) to indicate that this is the fraction of all cats …neutered at any given time compared to the annual neutering effort. That concept is confusing for some.
Line 145: I think it might be helpful to support the statement if the actual equations with the numbers the authors used were included here.
Line 151: this has the first p-value and it would be very helpful to include in the methods that the symbol is P. And in this sentence I’d add a few words to help the reader jump to “oh, this is a p-value and measure of statistical significance”. I would also like to see the regression equations on this page included in the manuscript. Maybe in some of the figure legends where the regression lines are shown? But also, the regression lines calculated for the Foley data could be in a table and compared to the new data. Removes a bit of the “black box” behind the scenes calculations and makes things even more transparent.
Love the lines for x and y intercept in figure 2!
Figure 3: are these monthly data? Can you clarify that to make it clear how this graph does or does not line up with the Foley graph of pregnancies?
Line 201: is this the x-intercept actually, not y?
Line 205-7: do you want to include here some additional data on the actual numbers of cats in residence? This is mentioned earlier and later but seems kind of an important juxtaposition.
Line 224-5: this is the definition of a median. Is there a specific reason for this statement?
Figure 4: can you indicate which points were outliers and therefore removed in the graphic for clarity?
Table 2: legend: please make clear that these are the calculated (for this manuscript) data points and not the ones directly from the prior publication.
Line 293: I don’t see any emphasis. Please double check.
Author Response
Thank you for your feedback—please see the attached file for our response.

Reviewer 2 Report
This article is an important re-evaluation of previous critiques on trap-neuter return programmes, and as such I recommend publication with some substantive, but mainly editorial/structural revisions.
However, as to its structure, I think the manuscript would benefit from a better structure. Some aspects of the Methodology are not sufficiently clearly explained, or are explained in the Results or even the Discussion. The Discussion introduces additional approaches and findings that have not been mentioned clearly in either the Methods or the Results. This may stem from the fact that the authors wanted to follow directly the approach by the article they criticise, but I think their work goes beyond Foley et al.'s approach and therefore they should not hide their additional analyses in the Discussion. This makes it much harder for the reader to actually see the full picture clearly. Some of their writing assumes too much that the reader is going to take the effort to also read Foley et al. simultaneously. The author's article should be fully understandable without having to refer back to the other article.
"Major" comments
L91ff: Provide a brief summary of the approach by Foley et al., including for example where their study took place. Do not assume that a reader of your article wants to take the effort to download Foley et al.'s article to read both side by side.
L135-137: Would it be possible to provide a table with the actual values, rather then to send the reader to go to the original article? Also "the methods described by Foley…" should at least be briefly outlined. Ideally all this should be in the main body of the article, but in the worst case (if there are word/page limit constraints imposed by the publisher) it could at least be presented in an electronic supplement.
L141: Do you mean Figure 1 in Foley et al. or do you mean your own Fig. 1? In that case perhaps write "Fig.1 in [19]". Your own Fig. 1 does not seem to be relating to the data in Foley. The whole sentence is very vague and does not allow the reader to understand what values were used in which way. Furthermore, it rather sounds like Methods than Results.
L145: Again: you seem to refer to Figures published by Foley, so perhaps state that more clearly as "Fig. 1a vs. 5a in Foley" (or something similar). The reader of your article does not have these figures at hand, so they cannot really follow your argument. I can confirm that I checked the statement, and while it is a bit difficult to see, the values doe indeed not completely match up. In fact, there are fewer data points in Foley's fig. 5a (N=8) than in Fig 1a (N=12) and 1b (N=7) vs. 5b (N=5).
For the values in brackets, it is not clear which of the two values within a bracket are their and which are your values.
L209-212: You lost me there: How were these values calculated or where did you obtain them from? They seem to be related to Table 1 in Foley et al., but how they got them is also not clear. If you could perhaps explain that a bit more clearly. L218-220 could be the explanation – but why would the explanation of the approach come after the result is stated? This is all quite confusing (both in Foley et al. and here).
L231: You mention in your Introduction and in the Results some simulation you run – but you do not explain in the Methods how this simulation was run. Please provide more details in the Methods.
L252/253: Could you a) state in the Results more explicitly that the relationship is not linear and b) perhaps also conduct an appropriate analysis to compare to?
L258/25): You mention here for the first time the term "critical annual neutering rate". You do not seem to define that in the Methods, nor explain how this was obtained. Please add that information in the Methods.
L290ff: You state clearly that long term data are needed to get a full picture for the regression analysis. Your own data exceed this value. To make the findings comparable (and to possibly strengthen your point that short-term data are not reliable), could you perhaps also analyse your own data only for the same time-span used by Foley et al.?
Fig. 6: this belongs in my opinion in the Results – compare also my comment to L 252/253.
L308ff and Fig. 7: Again, that should go into the Results, and the approach should also be explained in the Methods.
Minor/editorial comments:
L25/L147 and elsewhere: replace "don't" with "do not" and "doesn't" with "does not" (In general, avoid contractions.)
L106: Perhaps better than "N1 is the population size at time 1" write "Nt is the population size at time t" (to be in line with the actual formula – it is understandable in any case). [I notice that Foley et al. have the same phrasing as you used.]
L120: free-roaming cat [not cats] populations
L147/148: Again, state more clearly whether you refer to Foley's or your figures.
Fig. 4: What are the red vs. the blue best fit lines? Explain more clearly in the caption.
L237: Perhaps mention in the Table header that the values for San Diego and Alachua follow the values use by Foley et al.
L251: Instead of vaguely referring to a "significant relationship" state more explicit whether it was a positive or negative relationship.
L254/Table 2/L299-300, L313 etc.: Instead of or additionally to the p-value, could you provide the R value? That is more informative than the p-value. Likewise, in Table 2, could you perhaps add the R values, so that the reader does not only see whether there is a positive or negative relationship, but also gets an indication of the strength. In regression analysis, the p-value very much depends on the sample size – the biologically more important aspect, however, is the effect size, i.e. the R-value.
L293/204: You write "emphasis added" but there is no emphasis added.
Fig. 7: Explain what the red and blue fit lines represent. (I think here and in some of the previous figures, the red lines are simply the zero line (but not in all cases) – personally, I find that confusing without further explanation.
L331: "and therefore ARE not included"
L335: citation needed (here as well, not just a few lines further below)
L395-397: Do you have a reference for this statement?
L453-457: Reference needed for statement. Given your catchy title, I think some brief reference back to fish and why the Ricker model may be relevant there but not cats could strengthen the main message (and support the choice of the title).
L551: page numbers missing
Additional comment: Not sure whether it is relevant: Do you see a problem with the approach by Foley to estimate free roaming cat populations by multiplying the proportion of households feeding feral cats with the number of cats they feed on average? Does that in any way take into account that different households could feed the same cats, i.e. instead of being a "minimum estimate", could this grossly overestimate the feral population? Would that affect the calculations?
Author Response

(The authors gave the same response as above.)
